# Accuracy and Reproducibility of Contrast-Enhanced Mammography in the Assessment of Response to Neoadjuvant Chemotherapy in Breast Cancer Patients with Calcifications in the Tumor Bed

**DOI:** 10.3390/diagnostics11030435

**Published:** 2021-03-04

**Authors:** Valentina Iotti, Moira Ragazzi, Giulia Besutti, Vanessa Marchesi, Sara Ravaioli, Giuseppe Falco, Saverio Coiro, Alessandra Bisagni, Elisa Gasparini, Paolo Giorgi Rossi, Rita Vacondio, Pierpaolo Pattacini

**Affiliations:** 1Radiology Unit, Department of Imaging and Laboratory Medicine, Azienda USL—IRCCS di Reggio Emilia, 42122 Reggio Emilia, Italy; valentina.iotti@ausl.re.it (V.I.); vanessa.marchesi@ausl.re.it (V.M.); sara.ravaioli@ausl.re.it (S.R.); rita.vacondio@ausl.re.it (R.V.); pierpaolo.pattacini@ausl.re.it (P.P.); 2Pathology Unit, Azienda USL—IRCCS di Reggio Emilia, 42122 Reggio Emilia, Italy; moira.ragazzi@ausl.re.it (M.R.); alessandra.bisagni@ausl.re.it (A.B.); 3Clinical and Experimental Medicine PhD Program, University of Modena and Reggio Emilia, 41121 Modena, Italy; 4Breast Surgery Unit, Azienda USL—IRCCS di Reggio Emilia, 42122 Reggio Emilia, Italy; giuseppe.falco@ausl.re.it (G.F.); saverio.coiro@ausl.re.it (S.C.); 5Oncology Unit, Azienda USL—IRCCS di Reggio Emilia, 42122 Reggio Emilia, Italy; elisa.gasparini2@ausl.re.it; 6Epidemiology Unit, Azienda USL—IRCCS di Reggio Emilia, 42122 Reggio Emilia, Italy; paolo.giorgirossi@ausl.re.it

**Keywords:** breast cancer, calcifications, contrast-enhanced mammography, neoadjuvant chemotherapy, treatment monitoring

## Abstract

This study aimed to evaluate contrast-enhanced mammography (CEM) accuracy and reproducibility in the detection and measurement of residual tumor after neoadjuvant chemotherapy (NAC) in breast cancer (BC) patients with calcifications, using surgical specimen pathology as the reference. Pre- and post-NAC CEM images of 36 consecutive BC patients receiving NAC in 2012–2020, with calcifications in the tumor bed at diagnosis, were retrospectively reviewed by two radiologists; described were absence/presence and size of residual disease based on contrast enhancement (CE) only and CE plus calcifications. Twenty-eight patients (77.8%) had invasive and 5 (13.9%) in situ-only residual disease at surgical specimen pathology. Considering CE plus calcifications instead of CE only, CEM sensitivity for invasive residual tumor increased from 85.7% (95% CI = 67.3–96%) to 96.4% (95% CI = 81.7–99.9%) and specificity decreased from 5/8 (62.5%; 95% CI = 24.5–91.5%) to 1/8 (14.3%; 95% CI = 0.4–57.9%). For in situ-only residual disease, false negatives decreased from 3 to 0 and false positives increased from 1 to 2. CEM pathology concordance in residual disease measurement increased (R squared from 0.38 to 0.45); inter-reader concordance decreased (R squared from 0.79 to 0.66). Considering CE plus calcifications to evaluate NAC response in BC patients increases sensitivity in detection and accuracy in measurement of residual disease but increases false positives.

## 1. Introduction

Neoadjuvant chemotherapies (NAC) aim at reducing the proportion of non-surgically treatable breast cancers and decreasing the need for mastectomy and/or axillary lymph node dissection [1,2,3]. NAC requires imaging tools to accurately predict pathological response and, consequently, to guide surgical planning. The accuracy of imaging in defining response is influenced by several variables, including the heterogeneity of breast cancer subtypes [4,5,6,7] and the antiangiogenetic effect of drugs [8,9].

Some clinical trials have recently proposed avoiding surgery for those specific breast cancer molecular subtypes, i.e. triple negative and HER2-positive, known to be exceptional responders to NAC [10,11]. For a patient to be considered eligible for non-operative management of breast cancer after NAC, both the invasive and in situ components need to be eradicated because in situ disease may serve as a nidus for future recurrence [10]. The ductal carcinoma in situ (DCIS) component is frequently associated with pleomorphic and fine linear branching calcifications on mammography or area of enhancement on magnetic resonance imaging (MRI), on the latter especially when associated with high nuclear grade and comedonecrosis. The antiangiogenetic effect of chemotherapeutic drugs, however, tends to reduce the enhancement of the residual neoplastic component in the tumor bed after NAC, while calcifications almost never disappear completely. For this reason, patients with calcifications in the tumor bed depicted in mammography have a higher probability of being false negative in the assessment of residual disease after NAC with MRI, which is currently considered the gold standard imaging tool in treatment monitoring but technically unable to detect calcifications [8,12,13].

In treatment monitoring, contrast-enhanced mammography (CEM), a dual-energy mammographic system using an intravenous injection of iodine contrast media, is able both to provide morphological data (including the detection of pathological calcifications) and to image neovascularity [14,15,16,17]. In the detection of residual disease after NAC, a meta-analysis reported a pooled sensitivity with CEM of 80.7% (95% CI 65.5–90.2%) and a pooled specificity of 94.0% (95% CI = 78.3–98.6%) [18]. The literature investigating the role of CEM in the assessment of residual breast cancer following NAC continues to increase [19,20,21,22,23]; to the best of our knowledge, this is the first study that aims to evaluate the specific contribution of calcifications in this context. The implications of residual calcifications after NAC are still debated; since some calcifications may represent treated cancer with calcified or necrotic tissue and sloughed cells, their persistence in the tumor bed evaluated exclusively with mammography has not correlated with residual neoplasia [24,25,26,27,28,29,30]. Jochelson et al. [31] retrospectively described an improvement in the definition of residual disease and identification of breast-conserving therapy candidates when combining the mammographic assessment of residual calcification and enhancement in the tumor bed on MRI compared to the evaluation of MRI alone.

The primary objective of this study was to evaluate CEM accuracy in the detection and measurement of residual tumor after NAC in breast cancer patients with evidence of calcifications in the tumor bed, with pathological examination of the excised tumor bed as the reference standard. In particular, the goal was to compare CEM accuracy when evaluating contrast enhancement only and contrast enhancement in association with calcifications. Finally, we evaluated CEM reproducibility in the detection and measurement of residual tumor.

## 2. Materials and Methods

### 2.1. Study Design

This monocentric retrospective cross-sectional study was approved by the Area Vasta Emilia Nord (AVEN) Ethics Committee (protocol number 2020/0119203). The Ethics Committee, given the retrospective nature of the study, authorized the use of data without patients’ informed consent if all reasonable efforts had been made to contact that patient.

### 2.2. Setting and Study Population

All consecutive breast cancer patients followed by the provincial Breast Unit of Reggio Emilia who were treated with NAC from 2012 to 2020, who presented calcifications in the tumor bed at diagnosis, and who underwent CEM at diagnosis and after NAC, were included in this study. Fourteen of the cases included in this study were part of a previously published study [19]. Researchers conducting systematic reviews and meta-analyses can request separate tables describing cases included in the previous study.

### 2.3. Clinicopathological Data

Clinical data, including patients’ age, sex, clinical stage, NAC therapy and type of surgery, and tumor characteristics were obtained from the electronic medical records.

Pathological features included histotype and presence of in situ component at diagnosis, biological characteristics (hormone receptors, proliferative index evaluated by Ki67, and HER2 status), presence and size of residual invasive or in situ carcinoma, presence of calcifications, and pathological stage (ypTN) according to TNM 8^th^ edition.

To evaluate CEM accuracy the following features were considered as pathological reference standards: presence/absence of residual tumor on histological examination and residual tumor size (mm). The measurement of residual tumor size depended on the pathological pattern of the residual tumor, corresponding to a different shrinkage pattern following NAC: solitary nodule (concentric shrinkage) or multifocal lesions (patch-like shrinkage). Microscopic evaluation of the tumor’s major axis was used in cases of a solitary nodule, whereas the major axis of the whole pathological area macroscopically detected was used in cases of multifocal lesions. As this was a retrospective study, with no pathological macro-sections available, those patients for whom it was not possible to dimensionally retrace the extension of the residual tumor component (e.g., for multiple enlargements or neoplastic foci spread throughout the surgical specimen) were excluded from analyses on size (accuracy and concordance).

### 2.4. CEM Image Acquisition and Retrospective Review

CEM is a dual-energy mammographic system developed by GE Healthcare (Chalfont St-Giles, UK). After contrast administration, a low- and a high-energy image for each breast standard projection are acquired in quick succession while the breast remains compressed; the low-dose image obtained is comparable to a standard digital mammogram, and the post-processing recombined image enhances the distribution of the iodine contrast medium. The intravenous contrast agent is administered by a nurse, under the supervision of a radiologist, using a hand-held battery powered injector (Optistat, Covidien, Walpole, MA, USA). The contrast agent ioversol 350 mg/mL was used until September 2019; since then, iohexol 350 mg/mL has been used, both with a dose of 1.5 mL/kg of body weight. Other technical characteristics of CEM have been previously described [19].

Two independent radiologists, experts in breast imaging (V.M. and S.R.), evaluated the low-energy and recombined images of CEM of the single patient before and after NAC, blinded to the pathological results on surgical specimen (gold standard) but not blinded to the previous examinations. The two readers described:-the absence or presence of contrast enhancement (CE) on recombined images, defining the maximum dimension (mm) of CE before NAC (defining the tumor bed) and after NAC (defining the residual disease);-the maximum extension (mm) and characteristics (according to BIRADS lexicon) of calcifications on low-energy images before NAC (defining the tumor bed) and after NAC (defining the residual disease);-the maximum extension (mm) of the combined evaluation of pathological calcifications and enhancement before (defining the tumor bed) and after (defining the residual disease) NAC;-the radiological response after NAC according to both the combined evaluation and the evaluation of CE only:
○absence of residual disease (complete response (rCR)) or○the persistence of residual disease (partial response (rPR), stable disease, and progressive disease compared to the initial tumor bed).

When assessing radiological response according to the combined evaluation, calcifications were evaluated first in terms of persistence of malignant characteristics (fine pleomorphic, coarse heterogeneous, fine linear/branching, etc.), then in terms of any variation in extension (decreased, stable, or increased) [30].

A case representing how radiological response was assessed on CEM images is reported in Appendix A. Discordant cases were discussed in consensus with a third breast radiologist (V.I.) for the final decision on absence/presence of residual tumor.

### 2.5. Statistical Analyses

The number of true negatives, true positives, false negatives, and false positives are reported for CEM detection of overall residual tumor (invasive and/or in situ), residual invasive carcinoma (on the whole population), and in situ carcinoma (after excluding patients with residual invasive carcinoma) using pathological examination of the excised tumor bed as the reference standard. CEM sensitivity, specificity, positive predictive value (VPP), and negative predictive value (VPN), with respective 95% confidence intervals, were calculated for CE only and for the global judgment based on CE and calcifications. These estimates are only reported when respective denominators reached more than 5 cases; for smaller subgroups we report crude numbers of numerators and denominators only.

The concordance between CEM and pathological examination in terms of residual tumor size measurement is graphically depicted by means of linear regressions with respective R squared coefficient estimates for each CEM reader, considering CE only and CE + calcifications. R squared coefficients with respective *p* values adjusted for the non-independence of observations were also calculated for CE only and CE + calcifications compared to pathological examination when considering the total number of observations, e.g., two readings for each case.

The reproducibility of CEM in the detection of residual tumor after NAC was estimated by means of Cohen’s K inter-rater agreement, while inter-reader reproducibility for residual tumor size measurement was evaluated with linear regression with R squared estimate, both for CE only and for CE + calcifications.

## 3. Results

### 3.1. Population

From 2012 to 2020, 111 patients were treated with NAC for breast cancer in Reggio Emilia provincial hospitals, undergoing CEM before and after NAC. Of these patients, 36 had calcifications in the tumor bed and were included in this retrospective study. Before NAC, all patients had invasive carcinomas which showed CE on CEM.

The median age was 52 (44; 61) years. Other clinical characteristics of the included patients are reported in Table 1. Of the 36 patients included, 28 (77.8%) had residual invasive carcinoma at the pathological examination of the excised tumor bed, while 5 (13.9%) had only in situ residual carcinoma, and 3 (8.3%) had no invasive nor in situ residual tumor.

### 3.2. Accuracy of CEM in Detecting Residual Tumor

In detecting invasive residual tumor, CEM had a sensitivity of 85.7% (24/28) when considering contrast-enhancement (CE) only; sensitivity increased to 96.4% (27/28) when considering calcifications as well. However, specificity decreased from 62.5% (5/8) for CE only to 12.5% (1/8) for CE + calcifications (Table 2). In particular, when considering CE + calcifications compared to CE only, three false negatives were avoided but four false positives were introduced. When considering overall residual tumor (invasive and in situ), the difference in sensitivity was even more evident (78.8% (26/33) for CE only vs. 96.8% (32/33) for CE + calcifications) due to six false negatives being avoided, at the cost of one added false positive. Of the eight patients remaining after excluding patients with residual invasive carcinoma, when considering CE only, CEM resulted in three false negatives, which were all avoided when considering calcifications as well. On the other hand, false positives increased from 1 to 2 when considering CE + calcifications as opposed to CE only.

### 3.3. Analysis of Discordant Cases

The three pathological complete responders (ypT0) were all IDC grade 2 at preoperative biopsy, two of whom were HER2+ and one luminal B HER2+. This latter was the only case correctly assessed with CEM as complete responder, both when considering CE only and when considering CE + calcifications, since it showed no residual enhancement, and calcifications both decreased in extension and changed from pleomorphic to indeterminate. Of the other two cases, one showed some faint foci of enhancement in the tumor bed, and pleomorphic calcifications were stable after NAC, characteristics that were considered expression of residual disease both with CE only and with CE + calcifications. The other case did not present post-NAC enhancement, resulting in a true negative for CE-only judgement, but had stable pleomorphic calcifications still considered malignant and generating a false positive when considering CE + calcifications (Figure 1 and Appendix A).

The five patients with in situ-only residual tumor (ypTis) were all IDC: two grade 2 and three grade 3; one luminal B, one triple-negative, and three HER2+. All presented pleomorphic calcifications before NAC, four associated with mass enhancement and one with non-mass enhancement, and four with DCIS component diagnosed at breast biopsy. After NAC, the residual foci of enhancement in the tumor bed associated with a decreased number of calcifications (still pleomorphic) in two patients were described as an expression of residual disease both with CE only and with CE + calcifications. When considering only invasive residual tumor, these cases were false positives; when considering in situ residual tumor as well, they were true positives. The three remaining ypTis patients showed no residual enhancement but still had pleomorphic calcifications in the tumor bed, two stable and one slightly decreased; they were considered partial responders on the CE + calcifications evaluation, and complete responders when considering CE only.

Of the 28 patients with residual invasive component on the surgical specimen, only 1 (luminal B HER2+ with residual IDC + DCIS) was considered a false negative on both evaluations (CE only and CE + calcifications) due to the lack of residual enhancement and 5 cm of stable indeterminate calcifications (Appendix A). Three other patients showed no residual enhancement despite the invasive residual component (all IDC, two with concomitant DCIS component): they were all false negative when evaluated according to CE only, but as all presented pleomorphic residual calcifications, they were considered true positives based on CE + calcifications (Figure 2).

### 3.4. Concordance Between CEM and Pathology in the Measurement of Residual Tumor Size

As it was not possible to measure the residual tumor component in five patients, they were excluded from this analysis. Figure 3 depicts the association between the residual size provided by each of the two CEM readers when considering CE only and CE + calcifications and the histopathological measurement of residual tumor.

When considering all observations (two readings for each case), R squared increased from 0.3755 (adjusted *p* = 0.0008) when considering CE only to 0.447 (adjusted *p* = 0.0001) when considering CE + calcifications. In particular, the increased concordance in size between CEM and pathology when considering CE + calcifications reflects a decrease in underestimations, with a slighter increase in overestimations. When looking at the graphs (Figure 3), it seems that cases with an in situ component in the residual tumor bed were those that contributed the most to the decrease in underestimations when considering calcifications as well. The shrinkage pattern of enhancement also influenced the concordance in size measurement between CEM and pathology. Both for CE only and for CE + calcifications, cases with patchy shrinkage more frequently resulted in overestimation when compared with patients with concentric shrinkage.

### 3.5. CEM Reproducibility

For tumor detection, the concordance between the two readers was complete, with both readers indicating the presence of residual tumor in 27/36 patients based on CE only and in 34/36 patients based on CE + calcifications.

As depicted in Figure 4, the concordance in size measurement between the two readers slightly decreased when considering both CE and calcifications compared to CE only, with R squared decreasing from 0.788 to 0.656. Indeed, the two readers’ differing interpretation of calcifications (pleomorphic vs. indeterminate) in two patients was responsible for the lower concordance in size measurement when considering CE + calcifications compared to CE only (Figure 5).

## 4. Discussion

Considering the combination of CE + calcifications instead of CE only, CEM sensitivity in the detection of invasive residual tumor increased from 85.7% to 96.4%, at the cost of a decrease in specificity, from 62.5% to 12.5%, while inter-reader reproducibility for detection remained 100%. The concordance between residual tumor size estimated through pre-operative imaging and pathology on surgical specimen increased because underestimations in imaging decreased, while inter-reader reproducibility for size measurement slightly decreased.

### 4.1. Comparison with Previous Studies

These results are consistent with those obtained evaluating MRI enhancement combined with mammography for evaluating calcifications. In a retrospective study of 111 patients who underwent MRI before and after NAC, 60 of whom considered candidates for breast-conserving surgery, MRI combined with mammography was more accurate in predicting the possibility of breast conserving surgery (92% vs. 88%) than was MRI alone. In two cases, the visualization of calcifications increased the measurement of residual disease extent, leading to mastectomy, while such visualization in a third case resulted in a larger lumpectomy [31]); this is consistent with our result of decreased underestimations when adding calcifications to CE assessment.

The consistency between results obtained with CEM and those obtained combining MRI and mammography is not surprising, especially in a population with pleomorphic and fine linear branching calcifications at mammography, and consequently with a high prevalence of DCIS. Several studies have already shown that CEM accuracy is similar to that of MRI for the invasive component thanks to its ability to identify residual malignant tissue with high proliferative activity, like MRI [8,12,13]. The DCIS component, instead, is more easily detected through the analysis of calcification morphology in mammography or low-dose images in CEM. This could be of specific interest in patients with invasive disease and DCIS shown on the pre-treatment biopsy since this population is less likely than that without DCIS to achieve pCR (31% vs. 43%; *p* = 0.038) [4,5]. Lower response, together with faint enhancement observed in DCIS, makes diagnosing residual disease in women with DCIS component challenging. Adding the analysis of calcifications proved to increase sensitivity, but obviously could not impact positively on specificity. Indeed, the presence of a residual in situ component in the tumor bed contributed the most to the decrease in underestimations when considering the combined evaluation CE + calcifications.

Further, the evaluation of contrast enhancement after NAC can be challenging, especially when there is a patchy shrinkage pattern, which may impact the concordance between size measurement and pathology. Both for CE only and for CE + calcifications, patchy shrinkage more frequently resulted in overestimation when compared with patients with concentric shrinkage.

A large proportion of DCIS could also be why we observed very low specificity in our study. A previous meta-analysis [18] found a pooled specificity for invasive residual disease of over 90%. Firstly, the very small number of patients with no residual disease resulted in a very inaccurate estimate of specificity. However, even with larger patient samples, finding similar specificities in a population selected with our inclusion criteria and in unselected populations of patients undergoing NAC is unlikely. We can conclude that specificity in women with calcifications is probably lower than that in women without calcifications, but also that specificity will decrease when considering CE + calcifications compared to CE alone. Instead, based on our dataset, we cannot provide estimates of false positive rate, specificity, and PPV that can be generalized to other clinical settings.

### 4.2. Limitations

The study has some limitations which must be acknowledged.

First, the sample size was small, even if our center is one of the first to use CEM both in research studies (19) and in clinical practice in patients undergoing NAC. Given the small size of the study, and particularly the small proportion of women with complete response, our results need to be confirmed by larger studies.

Moreover, we lacked macro-sections. A histological correlation focusing on calcifications with macro-sections would increase our current understanding of false positive results in the whole surgical specimen. This would, in turn, contribute to improving CEM specificity and precision in measurements.

### 4.3. Implications for Practice and Research

Treatment monitoring after NAC is mostly used to plan surgical management, with the aim of reducing mastectomy and allowing breast conserving surgery without inducing an increase in reoperations. To assure this goal, imaging and pathology must be in agreement in terms of tumor measurement, especially to avoid gross underestimation of the size of the tissue to be excised. The added value of CEM is its precise definition of the tumor bed before NAC thanks to the direct integrated visualization of suspicious calcifications on the low-energy images and enhancement on the recombined images. Similarly, after NAC, CEM visualization of the tumor bed makes it possible to note changes in both calcifications and enhancement. This could assist the surgeon in opting for a minimally invasive surgery and would save time when compared to the common practice of adding MRI to mammography [32].

Another possible clinical utility of assessing response to NAC in the near future is to identify those women with complete response who could be referred to a watch-and-wait approach rather than to surgery [10,11].

Both for post-NAC surgical planning and for this latter possible indication of CEM, sensitivity, i.e., avoiding missing residual disease or underestimating the dimension of the residual disease, is crucial. Assessing response to NAC must also include the in situ component since all the neoplastic tissue, whether invasive or in situ, must be surgically removed, given the high probability of recurrence after incomplete DCIS resection. The presence of residual DCIS does not affect long-term outcomes but has clinical implications regarding both immediate surgical management (more extensive resections despite excellent response to NAC of the invasive component) and reoperation since these patients have worse event-free survival [10,25]. In this scenario, pre-surgical imaging after NAC should include a precise evaluation of calcifications instead of being limited to contrast enhancement due to the fundamental semeiotic contribution of mammographic imaging in depicting and defining the suspicious component.

Our results need to be confirmed by larger, possibly prospective multicenter studies that include an evaluation of both enhancement and calcifications; macro-section could possibly be used as the reference standard. Another goal we should focus on is to better characterize the morphological characteristics of calcifications and their clinical significance.

The decrease in specificity when considering calcifications is not a concern since all patients undergo surgery after NAC. Should the wait-and-watch approach become the standard treatment in selected cases, the first goal is to maintain high sensitivity. However, in that case, larger studies aiming at a better characterization of calcifications and evaluating the combined use of multiple imaging and molecular biomarkers may lead to an improvement in specificity without decreasing sensitivity, thereby increasing the number of women that can avoid surgical treatment.

## 5. Conclusions

Considering both CE and calcifications to evaluate NAC response in breast cancer patients increases sensitivity for residual invasive and in situ disease as well as accuracy in determining the dimension of residual lesion. However, it does increase the false positive rate. Further studies are needed which include the evaluation of both CE and calcifications in order to confirm our results and possibly to improve specificity by focusing on a better characterization of calcifications.

## Figures and Tables

**Figure 1 diagnostics-11-00435-f001:**
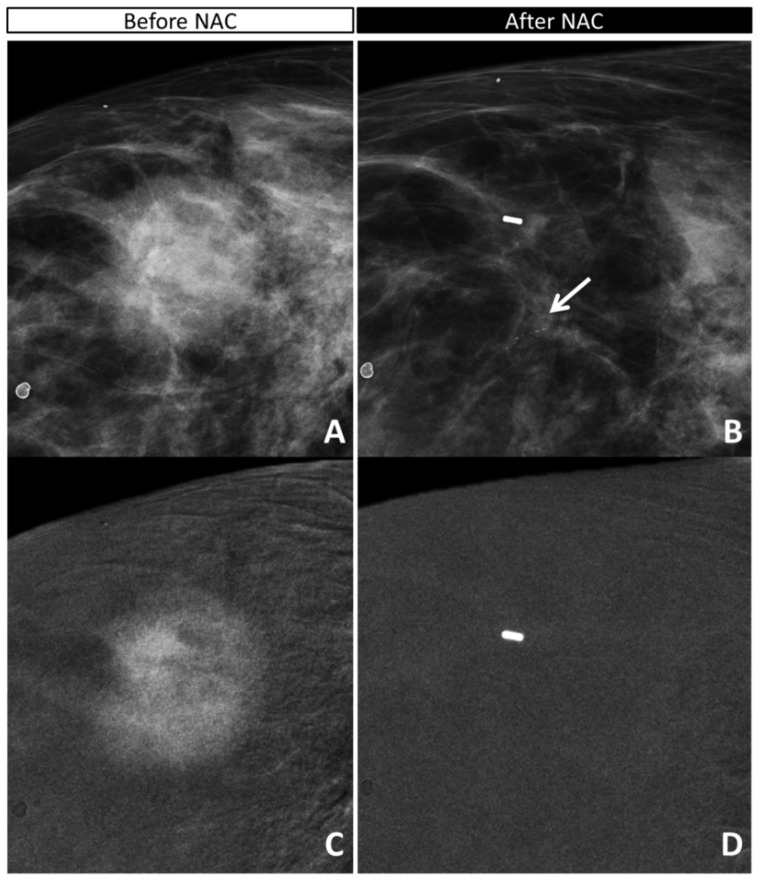
False positive for the combined evaluation and true negative for the CE evaluation: 67-yo woman with IDC G2 HER2+ with opacity and inner pleomorphic calcifications in the outer quadrant of the right breast (**A**,**C**: canio-caudal (CC) low-energy image). The in situ component was seen only on the surgical specimen. The opacity showed a strong mass enhancement before NAC (**C**: CC recombined image). After NAC (**B**: CC low-energy image; **D**: CC recombined image), the calcifications decreased slightly in size; remaining pleomorphic, they were considered pathological (**B**, arrow); no residual enhancement was visible surrounding the marker placed in the tumor bed, in the site of the previous opacity. The analysis of the surgical specimen revealed a complete response ypT0.

**Figure 2 diagnostics-11-00435-f002:**
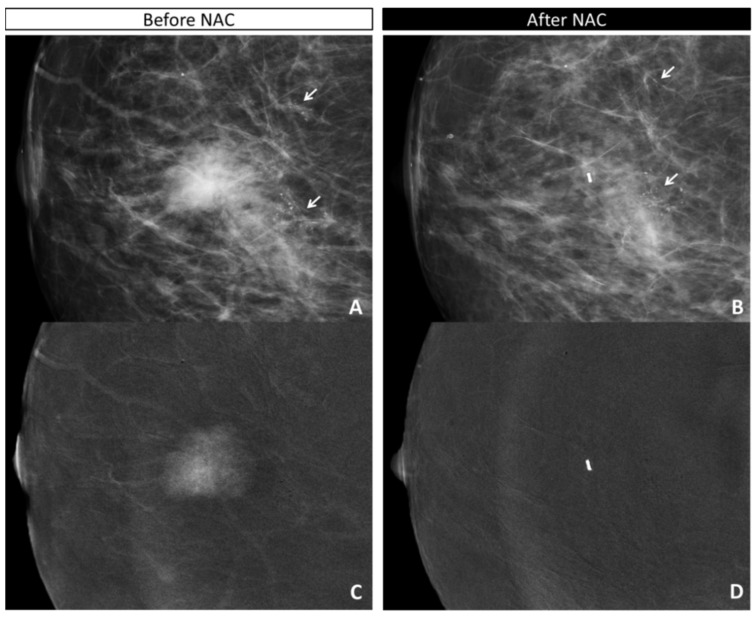
True positive for the combined evaluation and false negative on the CE-only evaluation: 55-yo woman with IDC G2 HER2+ with opacity and nearby, two clusters of pleomorphic calcifications in right breast (**A**,**C**: CC low-energy image; arrows). The in situ component was seen only on the surgical specimen. The opacity showed a strong mass enhancement before NAC (**C**: CC recombined image), while the clusters of calcifications showed no or only faint enhancement. After NAC (**B**: CC low-energy image; **D**: CC recombined image), the calcifications in both clusters decreased slightly (**B**, arrows); no residual enhancement was visible surrounding the marker placed in the tumor bed of the opacity. The analysis of the surgical specimen revealed 12 mm of residual IDC and multiple foci of DCIS (ypT1c).

**Figure 3 diagnostics-11-00435-f003:**
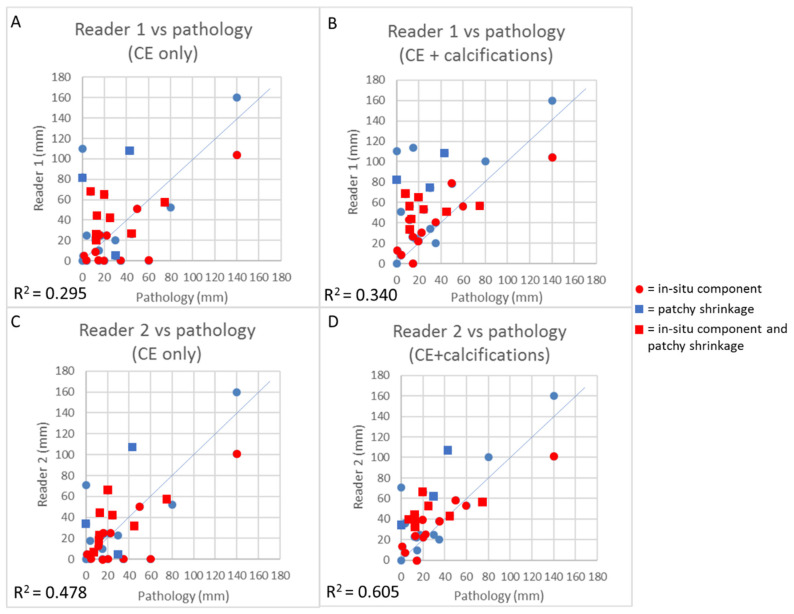
Linear regressions with respective R squared coefficients of CEM size measurement with pathology size measurement, respectively, for Reader 1 considering CE only (**A**), for Reader 1 considering CE + calcifications (**B**), for Reader 2 considering CE only (**C**), and for Reader 2 considering CE + calcifications (*n* = 31) (**D**). Squares represent cases with patchy shrinkage. Red dots and squares represent cases with in situ component in the residual tumor bed.

**Figure 4 diagnostics-11-00435-f004:**
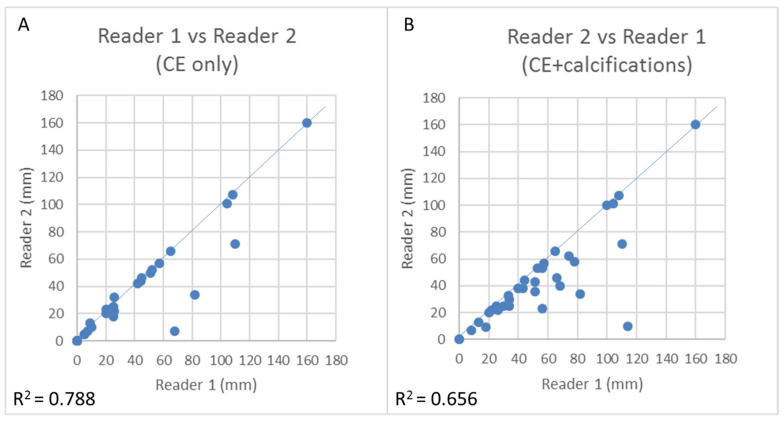
Linear regressions depicting concordance between the two CEM readers when considering CE only (**A**) and CE plus calcifications (**B**) (*n* = 36).

**Figure 5 diagnostics-11-00435-f005:**
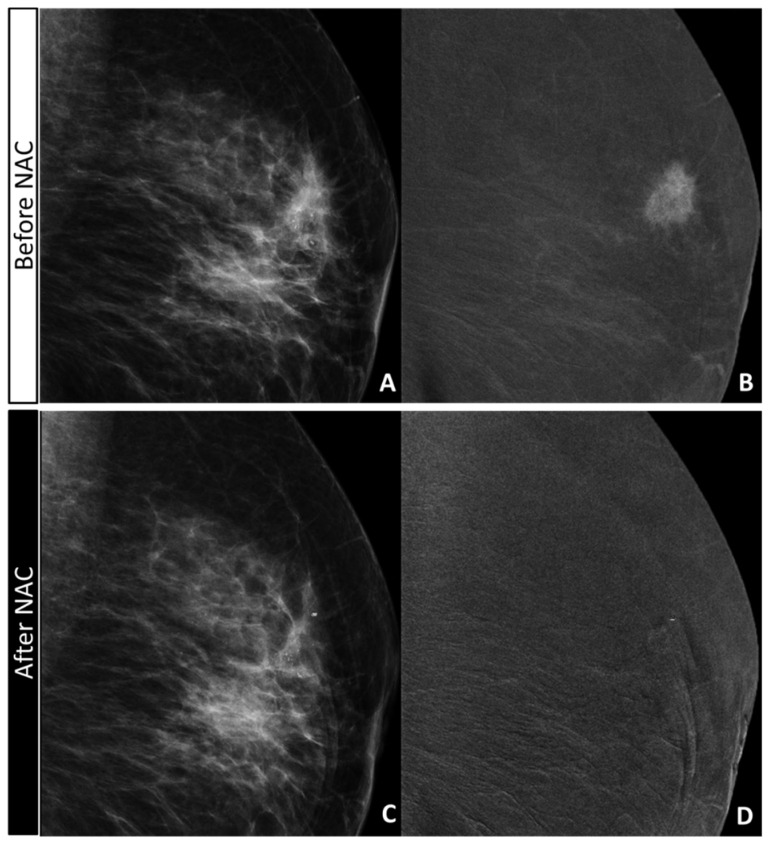
Inter-reader discordance in the evaluation of calcifications: 54-yo woman with IDC G3 HER2+ with opacity and nearby calcifications in left breast (**A**,**B**: medio-lateral-oblique (MLO) low-energy image). The in situ component was seen both on the initial biopsy and on the surgical specimen. For Reader 1, the calcifications after NAC remained pleomorphic and were considered as the measurement of the residual disease because more extensive (114 mm; **C**) than the concomitant faint residual rim enhancement (10 mm; **D**: MLO recombined image). For Reader 2, the same patient presented with a cluster of 15 mm of indeterminate calcifications after NAC (**C**), and thus considered CE only for the measurement of residual disease (10 mm; **D**). The analysis of surgical specimens reveled 15 mm of IDC and DCIS, with a better concordance for Reader 2 vs. Reader 1.

**Table 1 diagnostics-11-00435-t001:** Characteristics of the study population as a whole, only in patients with residual invasive tumor, and only in patients with in situ-only residual tumor.

	Overall (*n* = 36)	Invasive Residue (*n* = 28)	In Situ Residue (*n* = 5)
Mean age in years (IQR)	52 (44;61)	52 (43;58)	52 (44;60)
Pre-NAC tumor size; median (IQR) (mm)	50 (40;86.5)	50 (40;86.5)	46 (40;80.3)
cT2	23 (64%)	18 (64%)	4 (80%)
cT3	13 (36%)	10 (36%)	1 (20%)
**Histological Subtypes**			
IDC	34 (94%)	26 (92.8%)	5 (100%)
ILC	1 (3%)	1 (3.6%)	0
Metaplastic Carcinoma	1 (3%)	1 (3.6%)	0
**Ductal in situ component** (from pathology report)			
DCIS at diagnosis	16 (44%)	10 (36%)	4 (80%)
DCIS in the surgical specimen *	24 (67%)	18 (64%)	5 (100%)
DCIS disappeared after NAC	2 (/16 = 12%)	1 (/10 = 10%)	0
**Molecular Subtypes**			
Luminal B	10 (28%)	9 (32%)	1 (20%)
Luminal B HER2+	7 (19%)	6 (21%)	0
HER2+	13 (36%)	8 (29%)	3 (60%)
Triple negative	6 (17%)	5 (18%)	1 (20%)
Ki67, median (IQR)	30 (23.5–40)	30 (23.5–42.5)	30 (25–40)
**Calcifications pre-NAC**			
Indistinct	8 (22%)	7 (25%)	0
Pleomorphic	26 (72%)	19 (68%)	5 (100%)
Linear / branching	2 (6%)	2 (7%)	0
**Variation of calcifications after NAC**			
Decreased	12 (33%)	8 (29%)	3 (60%)
Stable	18 (50%)	14 (50%)	2 (40%)
Increased	6 (17%)	6 (21%)	0
**Enhancement after NAC**			
Concentric shrinkage	14 (39%)	14 (50%)	0
Patchy shrinkage	13 (36%)	10 (36%)	2 (20%)
No residual enhancement	9 (25%)	4 (14%)	3 (60%)
**ypT**			
0	3 (9%)		
is	5 (14%)		
1mic	2 (5%)		
1a	6 (16%)		
1a (m)	5 (/6=83%)		
1b	3 (9%)		
1b (m)	2 (/3 = 67%)		
1c	7 (19%)		
2	7 (19%)		
2 (m)	1 (/7 = 14%)		
3	3 (9%)		

IDC: invasive ductal carcinoma; ILC: invasive lobular carcinoma; DCIS: ductal carcinoma in situ; NAC: neoadjuvant chemotherapy; IQR, interquartile range; ypT: pathological post-neoadjuvant therapy T stage according to TNM 8th edition. * DCIS was detected only on the surgical specimen in 9/24 patients overall: 8/18 patients with invasive residue and 1/5 patients with in situ residue.

**Table 2 diagnostics-11-00435-t002:** Sensitivity, specificity, positive predictive value, and negative predictive value of CEM with respect to pathological reference standard in the detection of invasive residual tumor.

**Detection of Invasive Residual Tumor (*n* = 36)**								
	*TP*	*FP*	*TN*	*FN*	*Sensitivity* *(95% CI)*	*Specificity* *(95% CI)*	*PPV* *(95% CI)*	*NPV (95% CI)*
**CE only**	24	3	5	4	85.7% (67.3–96%)	62.5% (24.5–91.5%)	88.9% (76.4–95.2%)	55.6% (30.3–78.2%)
**CE+ calcs**	27	7	1	1	96.4% (81.7–99.9%)	12.5% (0.3–52.7%)	79.4% (74.6–83.5%)	
**Detection of Overall Residual Tumor (in situ or invasive) (*n* = 36)**								
	*TP*	*FP*	*TN*	*FN*	*Sensitivity* *(95% CI)*	*Specificity* *(95% CI)*	*PPV* *(95% CI)*	*NPV (95% CI)*
**CE only**	26	1	2	7	78.8% (61.1–91%)	66.7% (9.4–99.2%)	96.3% (83.9–99.2%)	22.2% (9.21–44.6%)
**CE+ calcs**	32	2	1	1	96.8% (84.2–99.9%)		94.1% (87.8–97.2%)	

CI, confidence interval; CE, contrast enhancement; calcs, calcifications.

## Data Availability

The data presented in this study are available on request from the corresponding author. The data are not publicly available due to ethical reasons.

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
