# Peer review of "Accuracy and Reproducibility of Contrast-Enhanced Mammography in the Assessment of Response to Neoadjuvant Chemotherapy in Breast Cancer Patients with Calcifications in the Tumor Bed"

_diagnostics, 2021, doi:10.3390/diagnostics11030435_

Round 1

Reviewer 1 Report

Summery : The purpose of this study was to evaluate  CEM accurancy and reproducibility in the detection and mesaurement of residual tumor in patient with breast cancer combined with cacification. The study group included 36 cases from 2012-2020. Reference standard was residual disease at surgical specimen pathology. The authors compare the performances of CEM considering CE only and CE+calcification. They conclude CE+calcification evaluation is more accurated than CE only with an increase of false positives. 

Strengths: Topic is timely and of interest to breast imaging radiologists. There is less literature about the contribution of calcifications in the assessment of residual breast cancer after NAC.

Weakness: small size of the study ; the reference standard (microscopic evaluation ) is not the better choice:  the results will be more significant with macro sectional measuraments; the statistical analyses could be enriched by more readers .

Author Response

Weakness: small size of the study, the reference standard (microscopic evaluation ) is not the better choice:  the results will be more significant with macro sectional measuraments; the statistical analyses could be enriched by more readers .

RE: We thank the Reviewer for the useful comments.

As to the sample size, we agree it is small. However, the inclusion criteria(presence of calcifications in the tumor bed in patients who underwent NAC and CEM) limited the number of eligible patients. Due to the extensive use of CEM in our center, where we also previously conducted a prospective study on CEM (19), our patient group is probably one of the largest in Italy as well as in Europe. This study is to be considered a pilot experience to open the way for prospective multicenter studies. We have better clarified these points in the Discussion. As to macro-sections, we completely agree, and have presented this point as a limitation. As to the last point, we do thank the Reviewer but adding a new reader would require a large amount of time to modify the paper.

We agree it was not easy to identify the study limitations in the previous version. We have now added a specific section.

The added paragraph is:

“3.2. Limitations

The study has some limitations which must be acknowledged.

First, the sample size is small, even if our center is one of the first to use CEM both in research studies (19) and in clinical practice in patients undergoing NAC. Given the small size of the study, and particularly the small proportion of women with complete response, our results need to be confirmed by larger studies.

Moreover, we lacked macro-sections. A histological correlation focusing on calcifications with macro-sections would increase our current understanding of false positive results in the whole surgical specimen. This would, in turn, contribute to improving CEM specificity and precision in measurements.”

Reviewer 2 Report

This study aimed to evaluate contrast-enhanced mammography (CEM) accuracy and re17 producibility in the detection and measurement of residual tumor after neoadjuvant chemotherapy 18 (NAC) in breast cancer (BC) patients with calcifications, using surgical specimen pathology as the reference. Pre- and post-NAC CEM images of 36 consecutive BC patients receiving NAC in 2012-2020, with calcifications in the tumor bed at diagnosis, were retrospectively reviewed by two radiologists; described were absence/presence and size of residual disease based on contrast enhancement (CE) only and CE plus calcifications. Twenty-eight patients (77.8%) had invasive and 5 (13.9%) in situ-only residual disease at surgical specimen pathology. Considering CE + calcifications instead 24 of CE only, CEM sensitivity for invasive residual tumor increased from 85.7% (95%CI=67.3-96%) to 96.4% (95%CI=81.7-99.9) and specificity decreased from 5/8 (62.5%; 95%CI=24.5-91.5%) to 1/8 (14.3%; 95%CI=0.4-57.9%). For in situ-only residual disease, false negatives decreased from 3 to 0 and false positives increased from 1 to 2. CEM pathology concordance in residual disease measurement in28 creased (R squared from 0.38 to 0.45); inter-reader concordance decreased (R squared from 0.79 to 0.66). Considering CE and calcifications to evaluate NAC response in BC patients increases sensitivity in detection and accuracy in measurement of residual disease but increases false positives. The paper is well written and clear. Some of the changes I would request are as follows: 1. In general it is better to have surgery as the tumor is removed. The authors could discuss how their technique can assist in minimally invasive surgery if there is residual cancer present after NAC. 2. The specificity seems to be low. Why is that? Is that due to inherent weakness in imaging. How can this be improved? Discussion on these lines would be helpful. 3. The conclusions are just two lines. I think it should be a paragraph discussing the work in a higher level. 4. Does the shrinkage of the tumor affect the specificity of the technique. 5. Can you compare the images before and after NAC and determine specificity in each case. Some discussions on these questions in the manuscript will make it stronger.

Author Response

Some of the changes I would request are as follows:
1. In general it is better to have surgery as the tumor is removed. The authors could discuss how their technique can assist in minimally invasive surgery if there is residual cancer present after NAC.

RE: We thank the Reviewer. In the section “Implications for practice” we have underlined this important point.

“This could assist the surgeon in opting for a minimally invasive surgery and would save time when compared to the common practice of adding MRI to mammography.”

  1. The specificity seems to be low. Why is that? Is that due to inherent weakness in imaging. How can this be improved? Discussion on these lines would be helpful.

Re: We thank the Reviewer for the observation. As we have underlined in the Discussion, the specificity may be affected by the inclusion criteria; we therefore state that our data are not generalizable.

However, we suppose that some improvement in specificity may be obtained with a better characterization of calcifications in the low energy images: to obtain data on the significance of different calcification characteristics, we need larger studies to gain more experience in the field.

Therefore, we have added:

“Our results need to be confirmed by larger, possibly prospective multicenter studies that include an evaluation of both enhancement and calcifications; macro-section could possibly be used as the reference standard. Another goal we should focus on is to better characterize the morphological characteristics of calcifications and their clinical significance.”

Moreover, we further discuss on improvement in specificity:

“The decrease in specificity when considering calcifications is not a concern, since all patients undergo surgery after NAC. Should the wait-and-watch approach become the standard treatment in selected cases, the first goal is to maintain a high sensitivity. However, in that case, larger studies aiming at a better characterization of calcifications and allowing an evaluation of the combined use of multiple imaging and molecular biomarkers may lead to an improvement in specificity without decreasing sensitivity, thereby increasing the number of women that can avoid surgical treatment.”

  1. The conclusions are just two lines. I think it should be a paragraph discussing the work in a higher level.

RE: We thank the Reviewer and we have added one more sentence:

“Further studies are needed which include the evaluation of both CE and calcifications in order to confirm our results and possibly to improve specificity by focusing on a better characterization of calcifications.”

  1. Does the shrinkage of the tumor affect the specificity of the technique.

RE: The 5 patients with pathologically patchy shrinkage were excluded from analyses regarding size measurement. All of these cases were classified as true positive in analyses on detection, so we may say that shrinkage pattern did not influence specificity in detection. The effect of shrinkage pattern on measurement is described in the Results and Discussion.

  1. Can you compare the images before and after NAC and determine specificity in each case.

RE: Before NAC, all included patients had invasive cancers, and all cancers had enhancement before NAC, so calculating specificity would not be possible while sensitivity would be 100%. We agree that a better description of pre-NAC tumor characteristics is useful. We have added that all cancers had enhancement pre-NAC (“Before NAC, all patients had invasive carcinomas which showed CE on CEM.”). We have also added T and tumor diameter at diagnosis in Table 1.

Some discussions on these questions in the manuscript will make it stronger.

Reviewer 3 Report

Comments for the manuscript titled: Accuracy and Reproducibility of Contrast-enhanced Mammography in the Assessment of Response to Neoadjuvant Chemotherapy in Breast Cancer Patients with Calcifications in the Tumor Bed

The authors evaluated the accuracy and reproducibility of CEM in detecting and measuring residual tumors in women with breast cancer with calcifications treated with NAC. Although the study provides a novel approach, the limited sample size for some of the study groups does not provide enough information to perform statistical analysis. In addition, the group shows that the approach of using CE + calcifications improves the sensitivity but greatly reduces the specificity on the detection of residual tumor.

Line 82-83: The authors state that the informed consent was collected whenever possible. I believe that the authors mean that all participants were consented different points in time, not that some were consented and other not. Please clarify.

Table 1: Include abbreviations on the footnote of the table. There is no statistical analysis to test for differences in proportions of the stratifications.

When indicating the individuals per groups use n=36 instead of n.36.

In the DCIS in situ component section of the table, clarify whether the DCIS at diagnosis was obtained from the pathology report. Clarify the difference between DCIS in the surgical specimen vs DCIS detected only on the surgical specimen. These categories should be clearly defined maybe on a footnote under the table because the reader can be confused.

In the molecular subtypes section, there is no Luminal A subtype. Please double-check since this is the most common breast cancer subtype.

Table 2: Include abbreviations on the footnote of the table. There is a reference to Grey boxes, but there are none on the table. Since there is not enough data to perform the statistical analysis of the women with in situ residual disease, that section of the table should be removed. The authors can include a statement in the text to explain that the analysis was performed but since the sample size is small and the statistical analysis could not be performed, it was not included on the Table.

Figures 4 and 5: On the figure captions, add the number of observations for each graph as (n=31).

Figure 5 is missing the R2 values on the graphs.

What are the possible causes and implications of the reduction of specificity from 62.5% to 12.5%? How do the authors think that this could be improved?

Author Response

The authors evaluated the accuracy and reproducibility of CEM in detecting and measuring residual tumors in women with breast cancer with calcifications treated with NAC. Although the study provides a novel approach, the limited sample size for some of the study groups does not provide enough information to perform statistical analysis. In addition, the group shows that the approach of using CE + calcifications improves the sensitivity but greatly reduces the specificity on the detection of residual tumor.
Line 82-83: The authors state that the informed consent was collected whenever possible. I believe that the authors mean that all participants were consented different points in time, not that some were consented and other not. Please clarify.

RE: We report the exact translation of the sentence of Ethics Committee waiving the mandatory collection of the informed consent “The Ethics Committee, given the retrospective nature of the study, authorized to use data without patients’ informed consent if all reasonable efforts have been made to contact that patient.”

Table 1: Include abbreviations on the footnote of the table. There is no statistical analysis to test for differences in proportions of the stratifications.

RE: We thank the Reviewer and we have included abbreviations in footnotes. As for the statistical analyses, as Table 1 is descriptive, outcomes are not reported; we therefore do not think it is necessary to add statistical tests. To define determinants of residual disease is beyond the scope of the study.

When indicating the individuals per groups use n=36 instead of n.36.

RE: Of course, thank you.

In the DCIS in situ component section of the table, clarify whether the DCIS at diagnosis was obtained from the pathology report. Clarify the difference between DCIS in the surgical specimen vs DCIS detected only on the surgical specimen. These categories should be clearly defined maybe on a footnote under the table because the reader can be confused.

RE: We thank the Reviewer for the comments, and we agree this section was not clear. We have improved the definitions in table 1.

In the molecular subtypes section, there is no Luminal A subtype. Please double-check since this is the most common breast cancer subtype.

RE: We have doublechecked and found no Luminal A subtype. Maybe this is because they are rarely treated with NAC; due to their low proliferation rate, they rarely respond to NAC.

Table 2: Include abbreviations on the footnote of the table. There is a reference to Grey boxes, but there are none on the table.

Since there is not enough data to perform the statistical analysis of the women with in situ residual disease, that section of the table should be removed. The authors can include a statement in the text to explain that the analysis was performed but since the sample size is small and the statistical analysis could not be performed, it was not included on the Table.

RE: We thank the Reviewer for the comments. We have added abbreviations in the footnotes and have removed both the reference to grey boxes and the section without statistical tests (the rates of FP and FN were already available in text).

Figures 4 and 5: On the figure captions, add the number of observations for each graph as (n=31).

RE: We have added the numbers.

Figure 5 is missing the R2 values on the graphs.

RE: We have added R2 values.

What are the possible causes and implications of the reduction of specificity from 62.5% to 12.5%? How do the authors think that this could be improved?

Re: Currently, as all patients are treated with surgery, we need to keep sensitivity high, while specificity is less important. Specificity becomes more important when the watch-and-wait approach is applied to selected cases. In this scenario, the decision not to surgically treat a woman may probably depend on the combination of more than one diagnostic test. Moreover, specificity may be improved by a better characterization of the morphological characteristics of calcifications.

We have added these sentences:

“The decrease in specificity when considering calcifications is not a concern, since all patients undergo surgery after NAC. Should the wait-and-watch approach become the standard treatment in selected cases, the first goal is to maintain high sensitivity. However, in that case, larger studies aiming at a better characterization of calcifications, and allowing an evaluation of the combined use of multiple imaging and molecular biomarkers may lead to an improvement in specificity without decreasing sensitivity, thereby increasing the number of women that can avoid surgical treatment.”

Round 2

Reviewer 2 Report

The authors have answered the criticisms from the previous review.